# SELF-SUPERVISED LEARNING FOR SEQUENTIAL RECOMMENDATION WITH MODEL AUGMENTATION

## ABSTRACT

The sequential recommendation aims at predicting the next items in user behaviors, which can be solved by characterizing item relationships in sequences. Due to the data sparsity and noise issues in sequences, a new self-supervised learning (SSL) paradigm is proposed to improve the performance, which employs contrastive learning between positive and negative views of sequences. However, existing methods all construct views by adopting augmentation from data perspectives, while we argue that 1) optimal data augmentation methods are hard to devise, 2) data augmentation methods destroy sequential correlations, and 3) data augmentation fails to incorporate comprehensive self-supervised signals. Therefore, we investigate the possibility of model augmentation to construct view pairs. We propose three levels of model augmentation methods: neuron masking, layer dropping, and encoder complementing. This work opens up a novel direction in constructing views for contrastive SSL. Experiments verify the efficacy of model augmentation for the SSL in the sequential recommendation.

## 1 INTRODUCTION

The sequential recommendation (Fan et al., 2021; Liu et al., 2021c; Chen et al., 2018; Tang & Wang, 2018; Zheng et al., 2019) aims at predicting future items in sequences, where the crucial part is to characterize item relationships in sequences. Recent developments in sequence modeling (Fan et al., 2021; Liu et al., 2021c) verify the superiority of Transform (Vaswani et al., 2017), i.e. the self-attention mechanism, in revealing item correlations in sequences. A Transformer (Kang & McAuley, 2018) is able to infer the sequence embedding at specified positions by weighted aggregation of item embeddings, where the weights are learned via self-attention. Existing works (Fan et al., 2021; Wu et al., 2020) further improve Transformer by incorporating additional complex signals.

However, the data sparsity issue (Liu et al., 2021c) and noise in sequences undermine the performance of a model in sequential recommendation. The former hinders performance due to insufficient training since the complex structure of a sequential model requires a dense corpus to be adequately trained. The latter also impedes the recommendation ability of a model because noisy item sequences are unable to reveal actual item correlations. To overcome both, a new contrastive self-supervised learning (SSL) paradigm (Liu et al., 2021b; Xie et al., 2020; Zhou et al., 2020) is proposed recently. This paradigm enhances the capacity of encoders by leveraging additional self-supervised signals. Specifically, the SSL paradigm constructs positive view pairs as two data augmentations from the same sequences (Xie et al., 2020), while negative pairs are augmentations from distinct sequences. Incorporating augmentations during training increases the amount of training data, thus alleviating the sparsity issue. And the contrastive loss (Chen et al., 2020) improves the robustness of the model, which endows a model with the ability to against noise.

Though being effective in enhancing sequential modeling, the data augmentation methods adopted in the existing SSL paradigm suffer from the following weaknesses:

- Optimal data augmentation methods are hard to devise. Current sequence augmentation methods adopts random sequence perturbations (Liu et al., 2021b; Xie et al., 2020), which includes crop, mask, reorder, substitute and insert operations. Though a random combination of those augmenting operations improves the performance, it is rather time-consuming to search the optimal

augmentation methods from a large number of potential combinations for different datasets (Liu et al., 2021b).

- Data augmentation methods destroy sequential correlations, leading to less confident positive pairs. The existing SSL paradigm requires injecting perturbations into the augmented views of sequences for contrastive learning. However, because the view construction process is not optimized to characterize sequential correlations, two views of one sequence may reveal distinct item relationships, which should not be recognized as positive pairs.

- Data augmentation fails to incorporate comprehensive self-supervised signals. Current data augmentation methods are designed based on heuristics, which already requires additional prior knowledge. Moreover, since the view construction process is not optimized with the encoder, data augmentation may only reveal partial self-supervised signals from data perspectives. Hence, we should consider other types of views besides data augmentation.

Therefore, we investigate the possibility of *model augmentation* to construct view pairs for contrastive learning, which functions as a complement to the data augmentation methods. We hypothesis that injecting perturbations into the encoder should enhance the self-supervised learning ability to existing paradigms. The reasons are threefold: Firstly, model augmentation is jointly trained with the optimization process, thus endows the end-to-end training fashion. As such, it is easy to discover the optimal view pairs for contrastive learning. Secondly, model augmentation constructs views without manipulation to the original data, which leads to high confidence of positive pairs. Last but not least, injecting perturbation into the encoder has distinct characteristics to data augmentation, which should be an important complement in constructing view pairs for existing self-supervised learning scheme (Liu et al., 2021b; Zhou et al., 2020).

This work studies the model augmentation for a self-supervised sequential recommendation from three levels: 1) neuron masking (dropout), which adopts the dropout layer to randomly mask partial neurons in a layer. By operating the dropout twice to one sequence, we can perturb the output of the embedding from this layer, which thus constructs two views from model augmentation perspective (Gao et al., 2021). 2) layer dropping. Compared with neuron masks, we randomly drop a complete layer in the encoder to inject more perturbations. By randomly dropping layers in an encoder twice, we construct two distinct views. Intuitively, layer-drop augmentation enforces the contrast between deep features and shallows features of the encoder. 3) encoder complementing, which leverages other encoders to generate sequence embeddings. Encoder complementing augmentation is able to fuse distinct sequential correlations revealed by different types of encoders. For example, RNN-based sequence encoder (Hidasi et al., 2015) can better characterize direct item transition relationships, while Transformer-based sequence encoder models position-wise sequential correlations. Though only investigating SSL for a sequential recommendation, we remark that model augmentation methods can also be applied in other SSL scenarios. The contributions are as follows:

- We propose a new contrastive SSL paradigm for sequential recommendation by constructing views from model augmentation, which is named as SRMA.

- We introduce three levels of model augmentation methods for constructing view pairs.

- We discuss the effectiveness and conduct a comprehensive study of model augmentations for the sequential recommendation.

- We investigate the efficacy of different variants of model augmentation.

## 2 RELATED WORK

### 2.1 SEQUENTIAL RECOMMENDATION

Sequential recommendation predicts future items in user sequences by encoding sequences while modeling item transition correlations (Rendle et al., 2010; Hidasi et al., 2015). Previously, Recurrent Neural Network (RNN) have been adapted to sequential recommendation (Hidasi et al., 2015; Wu et al., 2017), ostensibly modeling sequence-level item transitions. Hierarchical RNNs (Quadrana et al., 2017) incorporate personalization information. Moreover, both long-term and short-term item

transition correlations are modelled in LSTM (Wu et al., 2017) . Recently, the success of self-attention models (Vaswani et al., 2017; Devlin et al., 2018) promotes the prosperity of Transformer-based sequential recommendation models. SASRec (Kang & McAuley, 2018) is a pioneering work adapting Transformer to characterize complex item transition correlations. BERT4Rec (Sun et al., 2019) adopts the bidirectional Transformer layer to encode sequence. ASReP (Liu et al., 2021c) reversely pre-training a Transformer to augment short sequences and fine-tune it to predict the next-item in sequences. TGSRec (Fan et al., 2021) models temporal collaborative signals in sequences to recognize item relationships.

## 2.2 SELF-SUPERVISED LEARNING

Self-supervised learning (SSL) is proposed recently to describe "the machine predicts any parts of its input for any observed part"(Bengio et al., 2021), which stays within the narrow scope of unsupervised learning. To achieve the self-prediction, endeavors from various domains have developed different SSL schemes from either generative or contrastive perspectives (Liu et al., 2021a). For generative SSL, the masked language model is adopted in BERT (Devlin et al., 2018) to generate masked words in sentences. GPT-GNN (Hu et al., 2020) also generates masked edges to realize SSL. Other generative SSL paradigms in computer vision (Oord et al., 2016) are proposed. Compared with generative SSL, contrastive SSL schemes have demonstrated more promising performance. SimCLR (Chen et al., 2020) proposes simple contrastive learning between augmented views for images, which is rather effective in achieving SSL. GCC (Qiu et al., 2020) and GraphCL (You et al., 2020) adopts contrastive learning between views from corrupted graph structures. CL4SRec (Xie et al., 2020) and CoSeRec (Liu et al., 2021b) devise the sequence augmentation methods for SSL on sequential recommendation. This paper also investigates the contrastive SSL for a sequential recommendation. Instead of adopting the data augmentation for constructing views to contrast, we propose the model augmentation to generate contrastive views.

## 3 PRELIMINARY

### 3.1 PROBLEM FORMULATION

We denote user and item sets as $\mathcal{U}$ and $\mathcal{V}$ respectively. Each user $u \in \mathcal{U}$ is associated with a sequence of items in chronological order $s_u = [v_1, \ldots, v_t, \ldots, v_{|s_u|}]$, where $v_t \in \mathcal{V}$ denotes the item that $u$ has interacted with at time $t$ and $|s_u|$ is the total number of items. Sequential recommendation is formulated as follows:

$$\arg\max_{v_i \in \mathcal{V}} P(v_{|s_u|+1} = v_i \,|\, s_u), \tag{1}$$

where $v_{|s_u|+1}$ denotes the next item in sequence. Intuitively, we calculate the probability of all candidate items and recommend items with high probability scores.

### 3.2 SEQUENTIAL RECOMMENDATION FRAMEWORK

The core of a generic sequential recommendation framework is a sequence encoder $\mathsf{SeqEnc}(\cdot)$, which transforms item sequences to embeddings for scoring. We formulate the encoding step as:

$$\mathbf{h}_u = \mathsf{SeqEnc}(s_u), \tag{2}$$

where $\mathbf{h}_u$ denotes the sequence embedding of $s_u$. To be specific, if we adopt a Transformer (Kang & McAuley, 2018; Vaswani et al., 2017) as the encoder, $\mathbf{h}_u$ is a bag of embeddings, where at each position $t$, $\mathbf{h}_u^t$, represents a predicted next-item. We adopt the log-likelihood loss function to optimize the encoder for next-item prediction as follows:

$$\mathcal{L}_{\text{rec}}(u, t) = -\log(\sigma(\mathbf{h}_u^t \cdot \mathbf{e}_{v_{t+1}})) - \sum_{v_j \notin s_u} \log(1 - \sigma(\mathbf{h}_u^t \cdot \mathbf{e}_{v_j})), \tag{3}$$

where $\mathcal{L}_{\text{rec}}(u, t)$ denotes the loss score for the prediction at position $t$ in sequence $s_u$, $\sigma$ is the non-linear activation function, $\mathbf{e}_{v_{t+1}}$ denotes the embedding for item $v_{t+1}$, and $v_j$ is the sampled negative item for $s_u$. The embeddings of items are retrieved from the embedding layer in $\mathsf{SeqEnc}(\cdot)$, which is jointly optimized with other layers.

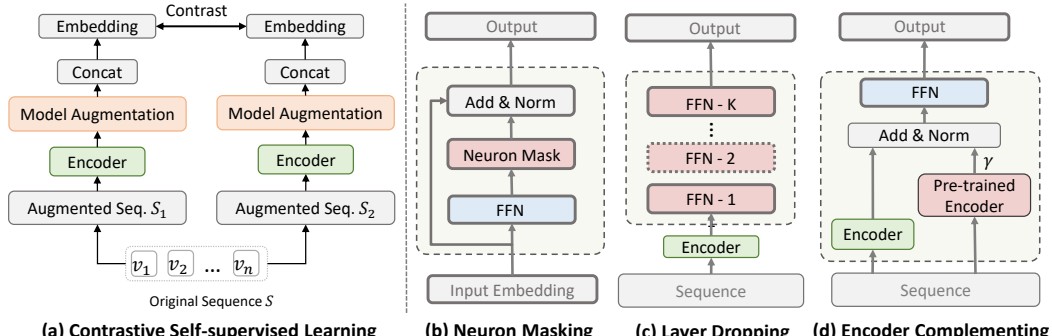

Figure 1: (a) The contrastive SSL framework with model augmentation. We apply the model augmentation to the encoder, which constructs two views for contrastive learning. (b) the neuron masking augmentation. We demonstrate the neuron masking for the Feed-Forward network. (c) the layer dropping augmentation. We add $K$ FFN layers after the encoder and randomly drop $M$ layers (dash blocks) during each batch of training. And (d) the encoder complementing augmentation. We pre-train another encoder for generating the embedding of sequences. The embedding from the pre-trained encoder is combined with the model encoder for contrastive learning.

### 3.3 CONTRASTIVE SELF-SUPERVISED LEARNING PARADIGM

Other than the next-item prediction, we can leverage other pretext tasks (Sun et al., 2019; Liu et al., 2021a;b) over the sequence to optimize the encoder, which harnesses the self-supervised signals within the sequence. This paper investigate the widely adopted contrastive SSL scheme (Liu et al., 2021b; Xie et al., 2020). This scheme constructs positive and negative view pairs from sequences, and employs the contrastive loss (Oord et al., 2018) to optimize the encoder. We formulate the SSL step as follows:

$$\mathcal{L}_{\mathrm{ssl}}(\tilde{\mathbf{h}}_{2u-1}, \tilde{\mathbf{h}}_{2u}) = -\log \frac{\exp(\mathrm{sim}(\tilde{\mathbf{h}}_{2u-1}, \tilde{\mathbf{h}}_{2u}))}{\sum_{m=1}^{2N} \mathbb{1}_{m \neq 2u-1} \exp(\mathrm{sim}(\tilde{\mathbf{h}}_{2u-1}, \tilde{\mathbf{h}}_{m}))}, \tag{4}$$

where $\tilde{\mathbf{h}}_{2u}$ and $\tilde{\mathbf{h}}_{2u-1}$ denotes two views constructed for sequence $s_u$. $\mathbb{1}$ is an indication function. $\mathrm{sim}(\cdot, \cdot)$ is the similarity function, *e.g.* dot-product. Since each sequence has two view, we have $2N$ samples in a batch with $N$ sequences for training. The nominator indicates the agreement maximization between a positive pair, while the denominator is interpreted as push away those negative pairs. Existing works apply data augmentation for sequences to construct views, *e.g.* Xie et al. (2020) propose *crop*, *mask*, and *reorder* a sequence and (Liu et al., 2021b) devises *insert* and *substitute* sequence augmentations. For sequential recommendation, since both SSL and the next-item prediction charaterize the item relationships in sequences, we add them together to optimize the encoder. Therefore, the final loss $\mathcal{L} = \mathcal{L}_{\mathrm{rec}} + \lambda \mathcal{L}_{\mathrm{ssl}}$. Compared with them, we adopt both the data augmentation and model augmentation to generate views for contrast. We demonstrate the contrastive SSL step in Figure 1(a).

## 4 MODEL AUGMENTATION

In this section, we introduce the model augmentation to construct views for sequences. We discuss three type of model augmentation methods, which are neuron mask, layer drop and encoder complement. We illustrate these augmentation methods in Figure 1(b), 1(c) and 1(d), respectively.

### 4.1 NEURON MASKING

This work adopts the Transformer as the sequence encoder, which passes the hidden embeddings to the next layer through a feed-forward network (FFN). During training, we randomly mask partial neurons in each FFN layer, which involves a masking probability $p$. The large value of $p$ leads to intensive embedding perturbations. As such, we generate a pair of views from one sequence from model perspectives. Besides, during each batch of training, the masked neurons are randomly

selected, which results in comprehensive contrastive learning on model augmentation. Note that, though we can utilize different probability values for distinct FFN layers, we enforce their neuron masking probability to be the same for simplicity. The neuron masking augmentation on FFN is shown in Figure 1(b). Additionally, we remark that the neuron mask can be applied to any neural layers in a model to inject more perturbations.

## 4.2 LAYER DROPPING

Dropping partial layers of a model decreases the depth and reduces complexity. Previous research argues that most recommender systems require only shallow embeddings for users and items (Dacrema et al., 2019). Therefore, it is reasonable to randomly drop a fraction of layers during training, which functions as a way of regularization. Additionally, existing works (Liu et al., 2020; He et al., 2016) claim that embeddings at shallow layers and deep layers are both important to reflect the comprehensive information of the data. Dropping layers enable contrastive learning between shallow embeddings and deep embeddings, thus being an enhancement of existing works that only contrasting between deep features.

On the other hand, dropping layers, especially those necessary layers in a model, may destroy original sequential correlations. Thus, views generating by dropping layers may not be a positive pair. To this end, instead of manipulating the original encoder, we stack $K$ FFN layers after the encoder and randomly drop $M$ of them during each batch of training, where $M < K$. We illustrate the layer dropping as in Figure 1(c), where we append $K$ additional FFN layers after the encoder and use dash blocks to denote the dropped layers.

## 4.3 ENCODER COMPLEMENTING

During self-supervised learning, we employ one encoder to generate embeddings of two views of one sequence. Though this encoder can be effective in revealing complex sequential correlations, contrasting on one single encoder may result in embedding collapse problems for self-supervised learning (Hua et al., 2021). Moreover, one single encoder is only able to reflect the item relationships from a unitary perspective. For example, the Transformer encoder adopts the attentive aggregation of item embeddings to infer sequence embedding, while an RNN structure (Hidasi et al., 2015) is more suitable in encoding direct item transitions. Therefore, contrasting between views from distinct encoders enables the model to learn comprehensive sequential relationships of items.

However, embeddings from two views of a sequence with distinct encoders lead to a non-Siamese paradigm for self-supervised learning, which is hard to train and suffers the embedding collapse problem (Koch et al., 2015; Chen & He, 2021). Additionally, if two distinct encoders reveal significantly diverse sequential correlations, the embeddings are far away from each other, thus being bad views for contrastive learning (Tian et al., 2020). Moreover, though we can optimize two encoders during a training phase, it is still problematic to combine them for the inference of sequence embeddings to conduct recommendations.

As a result, instead of contrastive learning with distinct encoders, we harness another pre-trained encoder as an encoder complementing model augmentation for the original encoder. To be more specific, we first pre-train another encoder with the next-item prediction target. Then, in the self-supervised training stage, we utilize this pre-trained encoder to generate another embedding for a view. After that, we add the view embeddings from a model encoder and the pre-trained encoder. We illustrate the encoder complementing augmentation in Figure 1(d). Note that we only apply this model augmentation in one branch of the SSL paradigm. And the embedding from the pre-trained encoder is re-scaled by a hyper-parameter $\gamma$ before adding to the embedding from the framework's encoder. The smaller value of $\gamma$ implies injecting fewer perturbations from a distinct encoder. And the parameters of this pre-trained encoder are fixed during training. Hence, there is no optimization for this pre-trained encoder and it is no longer required to take account of both encoders during the inference stage.

## 5 Experiments

### 5.1 Experimental Settings

**Dataset** We conduct experiments on three public datasets. Amazon Sports, Amazon Toys and Games (McAuley et al., 2015) and Yelp[1], which are Amazon review data in Sport and Toys categories, and a dataset for the business recommendation, respectively. We follow common practice in (Liu et al., 2021c; Xie et al., 2020) to only keep the '5-core' sequences. In total, Sports dataset has 35,598 users, 18,357 items and 296,337 interactions. Toys dataset contains 19,412 users, 11,924 items, and 167,597interactions. Yelp dataset consists 30,431 users, 20,033 items and 316,354 interactions.

**Evaluation Metrics** We follow (Wang et al., 2019; Krichene & Rendle, 2020; Liu et al., 2021c) to evaluate models' performances based on the whole item set without negative sampling and report standard *Hit Ratio*@$k$ (HR@$k$) and *Normalized Discounted Cumulative Gain*@$k$ (NDCG@$k$) on all datasets, where $k \in \{5, 10, 20\}$.

**Baselines** We include two groups of sequential models as baselines for comprehensive comparisons. The first group baselines are sequential models that use different deep neural architectures to encode sequences with a supervised objective. These include **GRU4Rec** (Hidasi et al., 2015) as an RNN-based method, **Caser** (Tang & Wang, 2018) as a CNN-based approach, and **SASRec** (Kang & McAuley, 2018) as one of the state-of-the-art Transformer based solution. The second group baselines additionally leverage SSL objective. **BERT4Rec** (Sun et al., 2019) employs a *Cloze* task (Taylor, 1953) as a generative self-supervised learning sigal. $\mathbf{S}^3\mathbf{Rec}$ (Zhou et al., 2020) uses contrastive SSL with 'mask' data augmentation to fuse correlation-ships among item, sub-sequence, and correspondinng attributes into the networks. We remove the components for fusing attributes for fair comparison. **CL4SRec** (Xie et al., 2020) maximize the agreements between two sequences augmentation, where the data augmentation are randomly selected from 'crop', 'reorder', and 'mask' data augmentations. **CoSeRec** (Liu et al., 2021b) improves the robustness of data augmentation under contrastive SSL framework by leveraging item-correlations.

**Implementation Details** The model encoder in SRMA is the basic Transformer-based encoder. We adopt the widely used SASRec encoder. The neuron masking probability is searched from $\{0.0, 0.1, 0.2, \ldots, 0.9\}$. For layer dropping, the $K$ is searched from $\{1, 2, 3, 4\}$, and $M$ is searched accordingly. As for encoder complementing, we search the re-scale hyper-parameter $\gamma$ from $\{0.005, 0.01, 0.05, 0.1, 0.5, 1.0\}$ and the pre-trained encoder is selected from a 1-layer Transformer and a GRU encoder.

### 5.2 Overall Performance

We compare the proposed paradigm SRMA to existing methods with respect to the performance on the sequential recommendation. Results are demonstrated in Table 1. We can observe that Transformer-based sequence encoders, such SASRec and BERT4Rec are better than GRU4Rec or Caser sequence encoders. Because of this, our proposed model SRMA also adopts the Transformer as sequence encoder. Moreover, the SSL paradigm can significantly improve performance. For example, the CL4SRec model, which adopts the random data augmentation, improves the performance of SASRec on HR and NDCG by 13.2% and 9.8% on average regarding the Sports dataset, respectively. Also, since SRMA enhances the SSL with both data augmentation and model augmentation, SRMA thus outperforms all other SSL sequential recommendation models. SRMA adopts the same data augmentation methods as CL4SRec. Nevertheless, SRMA significantly outperforms CL4SRec. On the sports dataset, we achieve 18.9% and 27.9% relative improvements on HR and NDCG, respectively. On the Yelp dataset, we achieve 4.5% and 6.4% relative improvements on HR and NDCG, respectively. And on Toys data, we achieve 8.6% and 13.8% relative improvements on HR and NDCG, respectively. In addition, SRMA also performs better than CoSeRec which leverages item correlations for data augmentation. Those results all verify the effectiveness of model augmentation in improving the SSL paradigm.

---

[1]https://www.yelp.com/dataset

Table 1: Performance comparisons of different methods. The best score is in bold in each row, and the second best is underlined.

| Dataset | Metric | GRU4Rec | Caser | SASRec | BERT4Rec | S³-Rec | CL4SRec | CoSeRec | SRMA |
|---------|--------|---------|-------|--------|----------|--------|---------|---------|------|
| Sports | HR@5 | 0.0162 | 0.0154 | 0.0206 | 0.0217 | 0.0121 | 0.0231 | 0.0287 | **0.0299** |
| | HR@10 | 0.0258 | 0.0261 | 0.0320 | 0.0359 | 0.0205 | 0.0369 | 0.0437 | **0.0447** |
| | HR@20 | 0.0421 | 0.0399 | 0.0497 | 0.0604 | 0.0344 | 0.0557 | 0.0635 | **0.0649** |
| | NDCG@5 | 0.0103 | 0.0114 | 0.0135 | 0.0143 | 0.0084 | 0.0146 | 0.0196 | **0.0199** |
| | NDCG@10 | 0.0142 | 0.0135 | 0.0172 | 0.019 | 0.0111 | 0.0191 | 0.0242 | **0.0246** |
| | NDCG@20 | 0.0186 | 0.0178 | 0.0216 | 0.0251 | 0.0146 | 0.0238 | 0.0292 | **0.0297** |
| Yelp | HR@5 | 0.0152 | 0.0142 | 0.0160 | 0.0196 | 0.0101 | 0.0227 | 0.0241 | **0.0243** |
| | HR@10 | 0.0248 | 0.0254 | 0.0260 | 0.0339 | 0.0176 | 0.0384 | 0.0395 | **0.0395** |
| | HR@20 | 0.0371 | 0.0406 | 0.0443 | 0.0564 | 0.0314 | 0.0623 | **0.0649** | 0.0646 |
| | NDCG@5 | 0.0091 | 0.008 | 0.0101 | 0.0121 | 0.0068 | 0.0143 | 0.0151 | **0.0154** |
| | NDCG@10 | 0.0124 | 0.0113 | 0.0133 | 0.0167 | 0.0092 | 0.0194 | 0.0205 | **0.0207** |
| | NDCG@20 | 0.0145 | 0.0156 | 0.0179 | 0.0223 | 0.0127 | 0.0254 | 0.0263 | **0.0266** |
| Toys | HR@5 | 0.0097 | 0.0166 | 0.0463 | 0.0274 | 0.0143 | 0.0525 | 0.0583 | **0.0598** |
| | HR@10 | 0.0176 | 0.0270 | 0.0675 | 0.0450 | 0.0094 | 0.0776 | 0.0812 | **0.0834** |
| | HR@20 | 0.0301 | 0.0420 | 0.0941 | 0.0688 | 0.0235 | 0.1084 | 0.1103 | **0.1132** |
| | NDCG@5 | 0.0059 | 0.0107 | 0.0306 | 0.0174 | 0.0123 | 0.0346 | 0.0399 | **0.0407** |
| | NDCG@10 | 0.0084 | 0.0141 | 0.0374 | 0.0231 | 0.0391 | 0.0428 | 0.0473 | **0.0484** |
| | NDCG@20 | 0.0116 | 0.0179 | 0.0441 | 0.0291 | 0.0162 | 0.0505 | 0.0547 | **0.0559** |

## 5.3 COMPARISON BETWEEN MODEL AND DATA AUGMENTATION

Because SRMA adopts the random sequence augmentation, we mainly focus on comparing with CL4SRec to justify the impacts of model augmentation and data augmentation. In fact, CL4SRec also implicitly uses the neuron masking model augmentation, where dropout layers are stacked within its original sequence encoder. To separate the joint effects of model and data augmentation, we create its variants 'CL4S. $p = 0$', which sets all the dropout ratios to be $0$, thus disables the neuron masking augmentation. Also, another variant 'CL4S. w/o D', which has no data augmentation are also compared. Additionally, we create two other variants of SRMA as 'SRMA w/o M' and 'SRMA w/o D' by disabling the model augmentation and data augmentation respectively. 'SRMA w/o M' has additional FFN layers compared with 'CL4S. $p = 0$'. The recommendation performance on the Sports and Toys dataset is presented in Table 2. We have the following observations. Firstly, we notice a significant performance drop of the variant 'CL4S. $p = 0$', which suggests that the neuron masking augmentation is rather crucial. It benefits both the regularization of the training encoder and model augmentation of SSL. Secondly, 'SRMA w/o D' outperforms other baselines on the Sports dataset and has comparable performance to 'CL4S.', which indicates the model augmentation is of more impact in the SSL paradigm compared with data augmentation. Thirdly, SRMA performs the best against all the variants. This result suggests that we should jointly employ the data augmentation and model augmentation in an SSL paradigm, which contributes to comprehensive contrastive self-supervised signals.

## 5.4 HYPER-PARAMETER SENSITIVITY

In this section, we vary the hyper-parameters in neuron masking and layer dropping to draw a detailed investigation of model augmentation.

**Effect of Neuron Masking.** Though all neural layers can apply the neuron masking augmentation, for simplicity, we only endow the FFN layer with the neuron masking augmentation and set the masking probability as $p$ for all FFN layers in the framework. We fix the settings of layer dropping and the encoder complementing model augmentation and select $p$ from $\{0.0, 0.1, 0.2, \ldots, 0.9\}$, where $0.0$ is equivalent to no neuron masking. Also, we compare SRMA with SASRec to justify the effectiveness of the SSL paradigm. The performance curves of HR@5 and NDCG@5 on the Sports and Toys dataset are demonstrated in Figure 2. We can observe that the performance improves first

Table 2: Performance comparison w.r.t. the variants of CL4SRec (CL4S.) and SRMA. M and D denote the model augmentation and data augmentation, respectively. $p = 0$ indicates no neuron masking. The best score in each column are in bold, where the second-best are underlined.

| Model | Sports | | | | Toys | | | |
|---|---|---|---|---|---|---|---|---|
| | HR | | NDCG | | HR | | NDCG | |
| | @5 | @10 | @5 | @10 | @5 | @10 | @5 | @10 |
| CL4S. w/o D | 0.0162 | 0.0268 | 0.0108 | 0.0142 | 0.0444 | 0.0619 | 0.0306 | 0.0363 |
| CL4S. $p = 0$ | 0.0177 | 0.0292 | 0.0119 | 0.0156 | 0.0451 | 0.0654 | 0.0305 | 0.037 |
| CL4S. | 0.0231 | 0.0369 | 0.0146 | 0.0191 | 0.0525 | 0.0776 | 0.0346 | 0.0428 |
| SRMA w/o D | 0.0285 | 0.0432 | 0.0187 | 0.0234 | 0.0504 | 0.0724 | 0.0331 | 0.0402 |
| SRMA w/o M | 0.0165 | 0.0272 | 0.0104 | 0.0138 | 0.0412 | 0.0590 | 0.0279 | 0.0336 |
| SRMA | **0.0299** | **0.0447** | **0.0199** | **0.0246** | **0.0598** | **0.0834** | **0.0407** | **0.0484** |

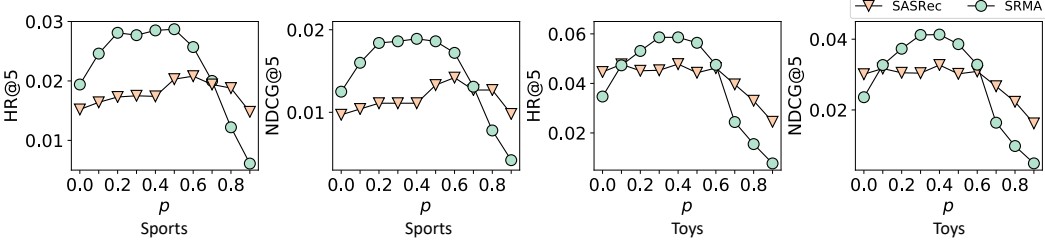

Figure 2: Performance comparison betweeen SASRec and SRMA in HR@5 and NDCG@5 w.r.t different values of neuron masking probability $p$ on Sports and Toys dataset.

and then drops when increasing $p$ from $0$ to $0.9$. The rising of the performance score implies that the neuron masking augmentation is effective in improving the ability of the sequence encoder for a recommendation. And the dropping of the performance indicates the intensity of model augmentation should not be overly aggressive, which may lead to less informative contrastive learning. As to SASRec, we recognize a higher score of SASRec when $p$ is large, which indicates the optimal model augmentation should be a slightly perturbation rather than a intensive distortion. Moreover, SRMA consistently outperforms SASRec when $0.1 < p < 0.6$. Since the only difference is that SASRec has no SSL module, we can thus concludes that the performance gains result from the contrastive SSL step by using the neuron masking.

**Effect of Layer Dropping.** The layer dropping model augmentation is controlled by two hyper-parameters, the number of additional FFN layers and the number of layers to drop during training, which is denoted as $K$ and $M$, respectively. Since we can only drop those additional layers, we have $M < K$. We select $K$ from $\{1, 2, 3, 4\}$ while $M$ are searched accordingly. Due to space limitation, we only report the NDCG@5 on the Sports and Toys dataset in Figure 3. We observe that $K = 2, M = 1$ achieves the best performance on both datasets, which implies the efficacy of layer dropping. Additionally, we also find that the performance on $K = 4$ is consistently worse than $K = 2$ on both datasets, which suggests that adding too many layers increases the complexity of the model, which is thus unable to enhance the SSL paradigm.

## 5.5 ANALYSES ON ENCODER COMPLEMENTING

In this section, we investigate the effects of encoder complementing augmentation for constructing views. Recall that we combine the embedding from the model's encoder and a distinct pre-trained encoder. For this complementary encoder, we select from a Transformer-based and a GRU-based encoder. Since the model's encoder is a 2-layer Transformer, this pre-trained encoder is a 1-layer Transformer to maintain diversity. We first pre-train the complementary encoder based on the next-item prediction task. As such, we empower the pre-trained encoder to characterize the sequential correlations of items. The comparison is conducted on both Sports and Toys datasets, which are shown in Table 3. The observations are as follows: Firstly, on the Sports dataset, pre-training a GRU encoder as a complement performs the best against the other two, which indicates that injecting dis-

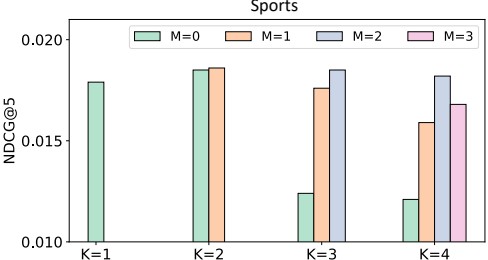 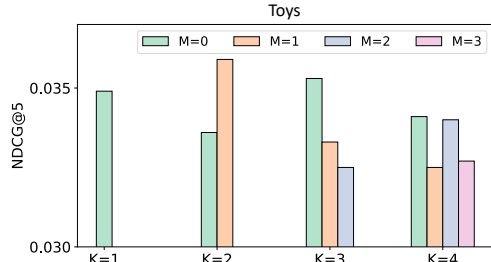

Figure 3: The NDCG@5 performance w.r.t. different $K$ and $M$ for layer dropping augmentation on Sports and Toys dataset.

Table 3: Performance comparison among SRMA without encoder complementing (w/o Enc.), with Transformer-based (-Trans) and with GRU-based (-GRU) complementary pre-trained encoder. The best score in each column is in bold.

| Encoders | Sports | | | | Toys | | | |
|---|---|---|---|---|---|---|---|---|
| | HR | | NDCG | | HR | | NDCG | |
| | @5 | @10 | @5 | @10 | @5 | @10 | @5 | @10 |
| w/o Enc. | 0.0269 | 0.0401 | 0.0181 | 0.0224 | 0.0567 | 0.0806 | 0.0389 | 0.0466 |
| -Trans | 0.0268 | 0.0408 | 0.0181 | 0.0226 | **0.0588** | **0.0811** | **0.0402** | **0.0474** |
| -GRU | **0.0281** | **0.0411** | **0.0186** | **0.0228** | 0.0577 | 0.0811 | 0.0395 | 0.047 |

tinct encoders for contrastive learning can enhance the SSL signals. Secondly, on the Toys dataset, adopting a 1-layer pre-trained Transformer as the complementary encoder yields the best scores on all metrics. Besides the effectiveness of encoder complementing, this result also suggests that the complementary encoder may not be overly different from the model's encoder on some datasets, which otherwise cannot enhance the comprehensive contrastive learning between views. Lastly, both Transformer-based and GRU-based pre-trained complementary encoders consistently outperform SRMA without encoder complementing, which directly indicates the necessity of encoder complementing as a way of model augmentation.

## 6 CONCLUSION

This work proposes a novel contrastive self-supervised learning paradigm, which simultaneously employs model augmentation and data augmentation to construct views for contrasting. We propose three-level model augmentation methods for this paradigm, which are neuron masking, layer dropping, and encoder complementing. We adopt this paradigm to the sequential recommendation problem and propose a new model SRMA. This model adopts both the random data augmentation of sequences and the corresponding three-level model augmentation for the sequence encoder. We conduct comprehensive experiments to verify the effectiveness of SRMA. The overall performance comparison justifies the advantage of contrastive SSL with model augmentation. Additionally, detailed investigation regarding the impacts of model augmentation and data augmentation in improving the performance are discussed. Moreover, ablation studies with respect to three-level model augmentation methods are implemented, which also demonstrate the superiority of the proposed model. Overall, this work opens up a new direction in constructing views from model augmentation. We believe model augmentation can enhance existing contrastive SSL paradigms with only data augmentation.

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
