# OpenReview forum: "Self-supervised Learning for Sequential Recommendation with Model Augmentation"
_ICLR.cc/2022/Conference — ICLR 2022 Submitted_

### Official Review · Reviewer_1XwB · 2021-10-27

**Correctness:** 3
**Technical Novelty And Significance:** 3
**Empirical Novelty And Significance:** 3
**Recommendation:** 5
**Confidence:** 4

**Main Review:**

Strengths:
1. The paper shows the effectiveness of model augmentation over data augmentation.
2. The paper shows that the effectiveness of model augmentation can help the model to achieve better performances.

Weakness:
1. The paper only considers model augmentation, but the model augmentation is closely related to model regularization. As shown in the paper "SSE-PT: Sequential Recommendation Via Personalized", regularization methods like SSE can help with the better performances too. Have the authors given much thoughts about the differences between model augmentation and model regularization? Are they similar concepts? If so, can we do some comparisons in experiments show the differences in terms of effectiveness?

**Summary Of The Paper:**

The paper is trying to address the sequential recommendation problem in which the goal is to predict the next items in user behavior. The paper proposes 3 levels of model augmentation methods: neuron masking, layer dropping and encoder complementing.

**Summary Of The Review:**

The paper addresses an important research problem and shows that a few model augmentation techniques can help with the sequential recommendation performances. My main concerns about the paper is the lack of explanation for the differences between model augmentation and regularization techniques. Addressing the concerns will help us understand us better if the proposed methods are general enough to be applied to other research problems.

---

> ### Author Response · Authors · 2021-11-20
> **Thanks for your comments!**
>
> SSE-PT is a good paper for sequential recommendation but may not related to our contrastive learning framework. But definitely, considering the model augmentation as a way of regularization is a very interesting point. Thanks for your suggestion! We will further extend our claims and conduct more experiments to support our claims.

---

### Official Review · Reviewer_D5iZ · 2021-10-28

**Correctness:** 3
**Technical Novelty And Significance:** 2
**Empirical Novelty And Significance:** 2
**Recommendation:** 3
**Confidence:** 4

**Main Review:**

Strengths:
1. This paper put forwards three kinds of model augmentation methods: neuron masking, layer dropping, and encoder complementing.
2. This work proposes an idea for constructing views for constrastive SSL.
Weaknesses:
1. What’s the motivation for using neuron masking/layer dropping/encoder complementing for model augmentation? Is there any theoretical analysis or intuition explanations?
2. Although the authors claim the method is model augmentation, I still consider that the method is a kind of data augmentation, because this paper only uses the three levels of model augmentation to construct view pairs for training the model.

**Summary Of The Paper:**

This paper proposes three levels of model augmentation methods: neuron masking, layer dropping, and encoder complementing. This work opens up a novel direction in constructing views for contrastive SSL and does experiments to verify the efficacy of model augmentation for the SSL in the sequential recommendation.

**Summary Of The Review:**

This paper proposes three levels of model augmentation methods: neuron masking, layer dropping, and encoder complementing. But the novelty and contributions are limited. Besides, it fails to explain the motivation of the proposed augmentation methods.

---

> ### Author Response · Authors · 2021-11-20
> **Thanks for your review.**
>
> Those three-level augmentations are based on the perturbation of model structures. We will conduct more experiments and claim the intuitions clearly later.

---

### Official Review · Reviewer_fdBG · 2021-11-02

**Correctness:** 3
**Technical Novelty And Significance:** 2
**Empirical Novelty And Significance:** 2
**Recommendation:** 3
**Confidence:** 4

**Main Review:**

### Strong points
1. two model augmentation methods for generating two views of a sequence.
2. The detailed experiments show the effectiveness of the proposed augmentation.
3. The paper is easy to read and follow

###  Weak points
1. overclaimed contributions. The authors should reorganize the contributions in the paper.
2. lack technical contributions. Dropping the overclaimed contributions, the contributions are marginal
3. lack significant test, since the improvements are very small, in the order of 0.001


**Summary Of The Paper:**

This paper  a new self-supervised learning (SSL) paradigm for sequence recommendation by  contrastive learning between positive and negative views of sequences based on model augmentation. The model augmentation methods includes neuron masking, layer dropping and encoder complementing. The proposed algorithm is evaluated with several real-world datasets, showing the efficacy of the proposed methods.



**Summary Of The Review:**

### Detailed comments

1. The neuron masking model augmentation has been proposed in CL4SRec, but claimed as a contribution in the paper. In this way, this model augmentation can not be also considered as a contribution.
2. Even though overclaimed contributions are included, the contributions may not sufficient. The proposed methods look a little straightforward and motivations are not supported by analysis.
3. The detailed setting of baselinesa are not given.
4. Regarding encoder complements, the authors should compare the baselines with GRU encoders as additional module. This is for illustrating the effectiveness of self supervised learning compared to simple ensemble .
5. "‘SRMA w/o D’ outperforms other baselines on the Sports dataset and has comparable performance to ‘CL4S.’, which indicates the model augmentation is of more impact in the SSL paradigm compared with data augmentation". The authors should discuss more about why the superority of SRMA w/o D to CL4S can indicate that the model augmentation is of more impact in the SSL paradigm compared with data augmentation.

---

> ### Author Response · Authors · 2021-11-20
> **Thanks for your comments!**
>
> Thanks for the comments! This paper is to highlight the possibility in model augmentation for contrastive learning. We recognize your suggestions and will conduct more experiments to support our claims.

---

### Decision · Program_Chairs · 2022-01-20

**Decision:**

Reject

**Comment:**

This paper proposed a self-supervised learning view for sequential recommendation with different forms of model augmentation: neuron masking, layer dropping, and encoder complementing. Overall the scores are negative. The reviewers raised concerns mostly around the motivation of the proposed approach (which wasn't fully supported by the experimental results) as well as the limited contribution (especially considering some of the augmentation strategies have been proposed in the past). One reviewer also brought out an interesting connection between model augmentation and model regularization. The authors responded that they will keep improving the paper and hopefully we will see a much improved version in the next submission.